# Arterial Stiffness Assessed by Cardio-Ankle Vascular Index

**DOI:** 10.3390/ijms20153664

**Published:** 2019-07-26

**Authors:** Takayuki Namba, Nobuyuki Masaki, Bonpei Takase, Takeshi Adachi

**Affiliations:** 1Department of Cardiology, National Defense Medical College, 3-2 Namiki, Tokorozawa, Saitama 359-8513, Japan; 2Department of Intensive Care Medicine2, National Defense Medical College, 3-2 Namiki, Tokorozawa, Saitama 359-8513, Japan

**Keywords:** arterial stiffness, heart failure, endothelial dysfunction, cardio-ankle vascular index

## Abstract

Arterial stiffness is an age-related disorder. In the medial layer of arteries, mechanical fracture due to fatigue failure for the pulsatile wall strain causes medial degeneration vascular remodeling. The alteration of extracellular matrix composition and arterial geometry result in structural arterial stiffness. Calcium deposition and other factors such as advanced glycation end product-mediated collagen cross-linking aggravate the structural arterial stiffness. On the other hand, endothelial dysfunction is a cause of arterial stiffness. The biological molecular mechanisms relating to aging are known to involve the progression of arterial stiffness. Arterial stiffness further applies stress on large arteries and also microcirculation. Therefore, it is closely related to adverse outcomes in cardiovascular and cerebrovascular system. Cardio-ankle vascular index (CAVI) is a promising diagnostic tool for evaluating arterial stiffness. The principle is based on stiffness parameter β, which is an index intended to assess the distensibility of carotid artery. Stiffness parameter β is a two-dimensional technique obtained from changes of arterial diameter by pulse in one section. CAVI applied the stiffness parameter β to all of the arterial segments between heart and ankle using pulse wave velocity. CAVI has been commercially available for a decade and the clinical data of its effectiveness has accumulated. The characteristics of CAVI differ from other physiological tests of arterial stiffness due to the independency from blood pressure at the time of examination. This review describes the pathophysiology of arterial stiffness and CAVI. Molecular mechanisms will also be covered.

## 1. Introduction

Arterial stiffness increases with age [1]. However, various co-factors with aging, such as lifestyle, metabolic disease, and renal failure, can accelerate its progression. Arterial stiffness augments pulse wave reflection by increasing peripheral arterial resistance and central arterial wall stiffness. Arterial stiffness increases cardiac afterload, which can cause left ventricular hypertrophy, diastolic dysfunction, and heart failure even though systolic function is preserved. Thus, inhibition, or slowdown, of the progression of arterial stiffness is an ideal way to prevent cardiovascular disease (CVD) and heart failure. 

For this purpose, appropriate evaluation for arterial stiffness is necessary. Cardio-ankle vascular index (CAVI) as an index of arterial stiffness was first proposed in 2006 [2]. CAVI improves the dependency of measurement value on blood pressure (BP) at measurement, which exists in brachial-ankle pulse wave velocity (baPWV). The principle of CAVI is based on the results of a previous experimental study of stiffness parameter β [3]; therefore, this is not an index of adjustment for BP. Because CAVI is not a simple measurement such as propagation of pulse, the effectiveness should be evaluated in clinical practice. This review aims to summarize the pathophysiology of arterial stiffness and the fundamentals of CAVI. We believe assessing the relationships between CAVI, biomarkers, and clinical prognosis of heart failure will lead to a better understanding of arterial stiffness.

## 2. The Mechanism of Arterial Stiffness

Arterial stiffness is characterized by impaired distensibility of large arteries. The structure of the arterial wall consists of three layers: intima, media, and adventitia. The possible mechanisms of arterial stiffness in each layer of arterial wall are shown in Figure 1. The media is sublayered into lamellar units defined by fenestrated sheets of elastic fibers. The number of lamellar units is up to 60 in larger elastic arteries such as thoracic aorta and decreases along the arterial tree [4]. The elastic fibers provide reversible elasticity to the arteries. However, mechanical fracture due to fatigue failure for the pulsatile wall strain induces vascular remodeling, which activates elastolytic enzymes including matrix-metalloproteinases (MMP) [5]. Four MMPs are known as elastases, gelatinase A (MMP-2), matrilysin (MMP-7), gelatinase B (MMP-9), and macrophage elastase (MMP-12). Vascular cells, as well as inflammatory cells such as macrophages and polymorphonuclear neutrophils, secret MMP-2, and MMP-9, which activate latent transforming growth factor-β [6]. On the other hand, increased luminal pressure, or hypertension, stimulates excessive collagen production [7]. Consequently, elastin fibers are replaced by stiffer collagen fibers with age. The reduced amounts of elastin in extracellular matrix composition and the subsequent alterations of arterial geometry result in loss of the integrity and material and/or structural arterial stiffness [8,9]. 

Endothelial dysfunction is considered to be a functional cause of arterial stiffness in addition to the medial degeneration. Vascular tone is determined by the balance of endothelium-derived vasoactive substances that include vasodilators such as nitric oxide (NO), prostaglandins, endothelium-derived hyperpolarization factor, epoxyeicosatrienoic acids, and vasoconstrictors such as angiotensin II, endothelin-1, prostanoids, and isoprostanes [10]. In them, NO is a physiologically relevant molecule to maintain vascular homeostasis. NO exerts anti-proliferative and anti-inflammatory effects on endothelium. NO promotes vasodilatation and prevents leukocyte adhesion [11], thrombocyte aggregation [12], and smooth muscle cell proliferation [13]. NO secreted from endothelium activates guanylate cyclase in vascular smooth muscle cells (VSMC), which increases cyclic guanosine monophosphate formation and reduces intracellular VSMC Ca^2+^ concentration leading to vascular relaxation. Therefore, deficiency of NO production caused by endothelial dysfunction increase peripheral arterial resistance, which promotes pulse wave reflection in the central arteries and causes additional stress on the vessel wall leading to intima–media thickening [14].

l-Arginine serves as the sole source of nitrogen for NO. Endothelial NO synthase (eNOS) converts l-arginine to l-citrulline with producing molecules of NO. The restoration of l-arginine from l-citrulline is catalyzed by the enzymes argininosuccinate synthase and argininosuccinate lyase in urea cycle [15]. l-asymmetric dimethylarginine (l-ADMA), a l-arginine derivative, increases with cardiovascular risks. The methylation of l-arginine is catalyzed by S-adenosylmethionine in methionine cycle. l-ADMA acts as a competitive NO synthase inhibitor and the relative increase of l-ADMA to l-arginine is associated with arterial stiffness [16]. Furthermore, NO synthesis may be influenced by global l-arginine bioavailability defined as the ratio of serum l-arginine to the sum of l-citrulline and l-ornithine [17]. The ratio is determined by the enzymes relating to l-arginine, especially eNOS, and arginase. Arginase is a rate-limiting enzyme in the urea cycle converting l-arginine to l-ornithine and urea. The enhanced activity of arginase induces chronic l-arginine deficiency and endothelial dysfunction. Arginase expression or activity is upregulated in hypertension, diabetes, and oxidative stress, as well as inflammation in human and animal models of various vascular diseases [18]. Moreover, l-ornithine is the precursor of polyamines that induce proliferation of vascular smooth muscle cells. Proline and hydroxyproline, which are the main precursors of collagens, are also synthesized from l-ornithine. Vascular endothelium-specific arginiase-1 knockout decreases arterial stiffening and fibrosis, elevated BP induced by high fat-high sucrose diet in mice [19]. Thus, an increase in l-arginine flux into the arginase pathway is involved in aortic stiffness via metabolic mechanisms.

Although the receptor of shear forces is not identified [20], increased mechanical force from blood flow directly stimulates the release of NO from the endothelium via phosphatidylinositol 3-kinase (PI3K)-Akt-mediated phosphorylation of eNOS [21,22,23]. Insulin stimulates the release of NO from endothelium by activating insulin receptor substrate (IRS)-1, which leads to PI3K-Akt mediated phosphorylation of eNOS. However, diabetes mellitus or metabolic disorders deteriorate the signal transition of IRS1-PI3K-Akt-eNOS pathway called insulin resistance. In addition, oxidative stress induces endothelial function by the oxidation of tetrahydrobiopterin (BH4) [24], an enzymatic cofactor of eNOS. Nutritional, therapeutic, and endothelium-derived factors including vitamin C, folate, and other antioxidants enhance endothelial BH4 bioavailability through chemical stabilization or scavenging of reactive oxygen species [25]. However, oxidative stress causes depletion of BH4 by oxidation into 7,8 dihydrobiopterin (BH2) in endothelial, so-called eNOS “uncoupling”. In this state, ONOO^−^, one of the reactive oxygen species, is produced instead of NO [26]. The deficiency of NO for these reasons accelerates the progression of arterial stiffness. The knockout of eNOS significantly increase arterial stiffness assessed by pulse wave velocity (PWV) of the aorta in animal models, suggesting the contribution of endothelial dysfunction to the development of arterial stiffness [27,28].

Sirtuin-1 (SIRT-1) is a nuclear nicotinamide adenine dinucleotide-dependent nuclear deacetylase [29] that acts on histone and nonhistone proteins, which regulate gene expression associated with aging. SIRT-1 promotes DNA damage repair, telomere stability [30], and exerts anti-inflammatory, antioxidant, and antiapoptotic effects on the endothelium [31,32]. Deregulated SIRT-1 is involved in vascular remodeling and arterial stiffness although a causative role remains to be established. The expression and activity of SIRT-1 in arteries decrease with aging [33]. SIRT-1 protects against high fat, high sucrose induced arterial stiffness [34], and caloric restriction activates SIRT-1 and increases eNOS expression [35]. A specific SIRT-1activator attenuates arterial stiffness induced by the deficiency of Klotho, an aging suppressor protein. Attenuated arterial stiffness is then assessed by PWV in mice [36]. Moreover, overexpression of SIRT-1 in endothelium increased circumferential cyclic strain of carotid and abdominal arteries and reduced nicotine-induced extracellular matrix remodeling and increase of PWV [37]. Thus, deficiency of SIRT-1 with age, metabolic disorders, and smoking can cause arterial stiffness.

There are other underlying mechanisms of arterial stiffening which may occur independently but relates to each other. Arterial stiffness has been considered to be related with lipids, calcification, fibrosis, chronic inflammation, advanced glycation end product-mediated collagen cross-linking, oxidative stress, and sympathetic nervous activity [38,39]. The traditional risk factors for CVD, for example, aging, hypertension, diabetes mellitus, smoking, and chronic kidney disease [40,41,42], impact arterial stiffness [43].

## 3. Physiological Examinations of Arterial Stiffness

There have been many non-invasive methods for validating arterial stiffness. The approaches of the methods can be mainly classified into the three groups; two-dimensional (2D) imaging technique, pulse wave analysis technique (PWA), and PWV. 

The approach of the 2D imaging technique is to measure the change of diameter, or sectional area under distensional pressures [44]. A representative index of the 2D imaging technique is arterial distensibility defined as the changes in the sectional area of a vessel divided by the difference of systolic and diastolic BPs [45]. The sectional areas are measured by various modalities, including ultrasound imaging, cardiac magnetic resonance imaging (CMR), and multi-detector row computed tomography (MDCT). PWA is the analysis of waveforms in the pressure tonometer, which enables an estimate of central hemodynamics and amount of augmented pressure resulted from superposition of forward and reflected pressure waves [46]. This type of method includes pulse pressure [47,48], central blood pressure (CBP) [49], augmentation index [50], and sub-endocardial viability ratio [51].

PWV is the velocity at which the pulse wave propagates through the circulatory system [52]. The part(s) of artery where the conduction time is measured determines the type of PWV, such as carotid-femoral PWV (cfPWV) [53], heart-femoral PWV (hfPWV) [54], and baPWV [55,56]. cfPWV is a golden standard, since cfPWV enables evaluation of only elastic arteries, whereas baPWV includes elastic arteries and muscular arteries. However, especially in Asian countries, baPWV is widespread for its convenience in clinical use. The placement of cuffs on upper arms and ankles can also obtain ankle brachial pressure index at the same time and save effort of taking clothes of patients off for the examination. baPWV has an increasing evidence for prediction of cardiovascular events. 

In addition, endothelial function tests are also important for patients suspected vascular disease because endothelial dysfunction causes functional arterial stiffening as described above. The endothelial function tests fundamentally assess the reactivity to hyperaemia after releasing the cuff at an upper arm or a forearm. This type of measurements includes venous occlusion digital plethysmography, flow-mediated dilation (FMD) of the brachial artery, and reactive hyperemia index (RHI) using fingertip peripheral arterial tonometry. However, the difference in the results of FMD and RHI was reported. This is supposed to be due to the difference of the targeted arteries [57,58]. 

CAVI is based on the principle of stiffness parameter β, an index of the 2D imaging technique, but requires PWV for the calculation. Therefore, reevaluation from clinical data is needed to know whether CAVI is superior to PWV for prediction of further cardiovascular events. The detail of the principle of CAVI is described in the following section.

## 4. Mechanical Principles of CAVI

CAVI was developed as a clinical test of arterial stiffness in Japan after the development of PWV. Previously, Hayashi et al. developed stiffness parameter β to stand for arterial stiffness in the 1980s [3]. The study described pressure-diameter hysteresis loop of arteries harvested from 18 hospital autopsies. The experimental apparatus equipped a pressure transducer and two pumps to make artificial pressure, and a displacement transducer to measure the changes of arterial diameter. A cylindrical segment of artery obtained from a patient was placed in a bath of Krebs-Ringer solution and intraluminal pressure was exerted gradually. The stiffness parameter β is a constant derived from the pressure-diameter hysteresis loop in a section of artery. They found that the constant is a characteristic value for the artery and can be used as an index of arterial stiffness. However, it is difficult to track the same section by ultrasound. For that reason, stiffness parameter β was usually used for aorta, carotid artery, and coronary artery in a research setting [59,60]. CAVI applied the stiffness parameter β to luminal volumetric changes and avoided determining a particular section of interest. This is a great advantage for common use. 

CAVI is calculated using BP and heart-ankle PWV (haPWV), monitoring of heart sounds and electrocardiogram. haPWV is calculated by dividing the distance from the aortic valve to the ankle artery by the sum of the time intervals between aortic valve closure sound and notch of the brachial pulse wave and between the rise of the brachial pulse wave and the ankle pulse wave. CAVI is derived from the stiffness parameter β and Bramwell-Hill formula [61] and determined using the following formula. The stiffness parameter β was determined by the following equation (Figure 2):Stiffness parameter β = ln(Ps/Pd) × D/ΔD(1)
where Ps and Pd are systolic and diastolic BP, respectively; D is the diameter of the artery, and ΔD is the change in D. Then, ΔD/D means strain.

The pressure-diameter curve of an artery is linearized by log-transformation of BP and a linear function is established between ln(Ps/Pd) and ΔD/D. The inclination of the straight line is defined as stiffness parameter β (ln(Ps/Pd)/(ΔD/D)).

On the other hand, the Bramwell-Hill formula was proven and derived in experiments. It is determined by the following formula:PWV^2^ = ΔP/ρ × V/ΔV(2)
where ΔP is pulse pressure, ρ is blood density of 1.05 g/ml, V is blood vessel volume, and ΔV is the change in V. V/ΔV can be approximated to following formula (ΔD^2^ is small and negligible):V/ΔV = (πL(D/2)^2^)/(πL( (D + ΔD)/2) ^2^ − πL(D/2)^2^)= D^2^/(2DΔD + ΔD^2^)≅ D/2ΔD(3)
where L is the length of the artery. Thus, V/ΔV in Bramwell-Hill formula can be replaced by D/2ΔD in stiffness parameter β. CAVI is defined as a leaner function of stiffness parameter β using the mean change of diameter in a segment where PWV is measured. CAVI is described as follows (Figure 3):CAVI = a (stiffness parameter β) + b= a [2(ρ/ΔP) × ln(Ps/Pd) × haPWV2] + b(4)
where Ps and Pd are systolic and diastolic BP, respectively; ΔP is pulse pressure (Ps − Pd); ρ is blood density of 1.05 g/mL; and a and b are constants. These constants are determined to fit a CAVI value to a value obtained by Hasegawa’s method [62,63,64].

This equation reflects the global stiffness of the aorta, femoral arteries, and tibial artery; CAVI represents the stiffness parameter β of a whole arterial segment [65]. CAVI correlates with the parameters of 2D imaging technique such as aortic distensibility in CMR [66] and aortic stiffness parameter β assessed by transesophageal echocardiography [67] and MDCT [68]. 

## 5. Strengthens and Limitation of CAVI

The most important characteristic of CAVI is the reproducibility due to the independence from BP at testing [69,70]. The ideal index of arterial stiffness is not thought to be influenced by the BP at measurement. CAVI is unchanged in the repetitive measurements at different systolic BP more than 10 mmHg [71]. In addition, several groups reported serial changes in CAVI during blood pressure by exercise in the same patient. A study demonstrated that CAVI is stable in the condition of elevated BP after stair climbing [72]. However, a recent study suggests that CAVI can be influenced by BP changes in response to handgrip as well as cfPWV and baPWV [73]. The study also shows the degree of correlation between changes of CAVI and BP depending on the type of pressor stimuli. 

The difference of drug-response between CAVI and baPWV was shown in a previous paper [74]. Metoprolol, β1-adrenoceptor blocker, did not change CAVI though systolic and diastolic pressure and baPWV were decreased. On the contrary doxazosin, α1-adrenoreceptor blocker, decreased systolic and diastolic pressures, CAVI, and baPWV together, suggesting doxazosin-induced arterial dilatation accompanying the reduction of peripheral vascular resistance. Therefore, CAVI value can be less influenced by change of BP shown in the metoprolol administration, but contraction of vascular smooth muscle of peripheral arteries as same as baPWV. 

CBP is a non-invasive test estimating ascending aortic BP and reflects direct stress on the heart. Therefore, CBP is thought to be a better indicator than brachial BP. CBP can also distinguish the different effects of anti-hypertensive drugs [75]. A previous study showed that CAVI is independent from CBP [76]. In the study, CAVI is closely related to left ventricular (LV) systolic and diastolic function measured by echocardiography, whereas CBP is more associated with LV hypertrophy.

The limitations of CAVI are mainly attributed to the placement of cuffs on ankles, which are equivalent to baPWV. For the CAVI evaluation, patients with severe aortic insufficiency, bilateral ankle-brachial index <0.9, or persistent, atrial fibrillation should be excluded because it is difficult to obtain accurate measurements in such patients [77]. Thus, patients with arteriosclerosis obliterans, the subgroup of the most severe arterial stiffness, have to be excluded from the analysis. 

## 6. Arterial Stiffness and Organ Damage

Clinically, increased arterial stiffness contributes to elevating left ventricular afterload and developing LV diastolic dysfunction. The Windkessel effect is when intermittent blood flow injected from the heart becomes a smoother and steadier stream after passing through the elastic aorta [78]. This is due to the transient expansion of the aorta at the time of transmitting pulse waves to peripheral arteries. This pulse wave is reflected at the bifurcation of the common iliac artery, whereas retrograde reflection pulse wave amplifies arterial BP when it returns to the ascending aorta. The return of the reflection pulse wave to the ascending aorta happens during diastole when PWV is slow. The Windkessel effect reduces cardiac afterload during systole and perfuse the coronary arteries during diastole. 

However, cardiac function can be deteriorated when the Windkessel effect has declined. The return of the reflection pulse wave occurs during late systole when PWV is fast due to arterial stiffness. The change in loading sequence amplifies systolic BP and increases pulse pressure [79,80,81] and, subsequently, increase LV afterload. This condition increases cardiac workload leading to increased myocardial oxygen demands and consequently promotes the development of myocardial hypertrophy and LV diastolic dysfunction [82]. On the other hand, the arrival of the reflection pulse wave during late systole causes a reduction in diastolic pressure. The hemodynamics leads to the reduction of coronary perfusion pressure and decreases cardiac function secondary to endomyocardial ischemia [83,84]. A previous experimental study, in which the blood flow of the aorta was switched into a bypass with a suffer Tyron conduit, demonstrated that the contractile function and efficiency of normal hearts are not altered by ejection into a stiff vascular system, but the myocardial O_2_ consumption for maintaining adequate flow is increased [85]. This model explains the pathogenesis of heart failure with preserved ejection fraction (HFpEF) caused by arterial stiffness, whereby there is little functional decrement of heart at rest but limit reserve capacity under conditions of increased demand.

Moreover, progression of large artery stiffness induces excessive penetration of pulsatility [86]. Aortic stiffness is much lower than muscular artery stiffness. The impedance mismatch produces proximal wave reflection at the aortic-muscular arterial interfaces. However, when aortic stiffness increases equivalent to muscular artery stiffness, the proximal wave reflections decreases, and the strong forward pulsatile power is transmitted directly to microcirculation. The pulsatile energy causes end-organ microvascular damage [87,88,89] and further elevates LV afterload. Especially, brain and kidney arteries are subjected to the pulsatile stress [90].

## 7. CAVI and Diagnosis of Diastolic Dysfunction and Heart Failure with Preserved Ejection Fraction

Approximately half of the patients with heart failure have preserved LV ejection fraction, called HFpEF [91,92,93]. The definition of HFpEF is defined as heart failure with ejection fraction greater than 50% [94,95,96]. HFpEF is a state of cardiac decompensation for elevated LV filling pressure in diastole although cardiac ability of contraction is not deteriorated. This phenomenon is thought to be significantly caused by distensibility of LV wall. Inversely, the detection of LV diastolic dysfunction is necessary for the diagnosis of HFpEF [97]. HFpEF is developed by aging, hypertension, diabetes mellitus, chronic kidney disease, metabolic syndrome, and arterial stiffness [98,99]. The progression of arterial stiffness in patients with heart failure was reported to correlate with poor prognosis; therefore, assessment and management of arterial stiffness are clearly imperative [100].

The diagnosis of diastolic dysfunction is achieved by abnormal left ventricular relaxation and increased left ventricular chamber stiffness. Cardiac catheterization is the gold standard for assessment of diastolic dysfunction [101]. Catheterization provides intracardiac pressures and LV diastolic properties but is difficult to perform routinely due to its invasiveness. Doppler echocardiography is advantageous for its non-invasiveness and feasibility in assessing diastolic dysfunction. The echo-examination provides the indexes of diastolic dysfunction and filling LV pressures, which closely correlates those obtained from pressure curves of catheterization [102]. Thus, the diastolic dysfunction is clinically diagnosed by echocardiography. The criteria of diastolic dysfunction include pattern and ratio of early to late diastolic transmitral flow velocity (E/A) and deceleration time (DT). LV filling index can be evaluated by ratio of transmitral flow velocity to mitral annular velocity (E/E’). However, these parameters cannot be separated and interrelated each other [103]. American Society of Echocardiography and the European Association of Cardiovascular Imaging recommended a diagnostic algorithms for diastolic dysfunction comprising four echocardiographic parameters; indexed left atrial volume (LAVI; >34 mL/m^2^), E’ (septal E’ < 7 cm/s, lateral E’ < 10 cm/s), E/E’ (>14) and tricuspid regurgitation velocity (>2.8 m/s) [104].

The close relationships between CAVI, diastolic dysfunction classified by E/A, and DT in Doppler echocardiography were described previously [105]. The relationship of CAVI and diastolic function should be evaluated separately by systolic cardiac function. CAVI can correlate negatively with E/A and positively with DT in patients of ischemic heart disease with preserved ejection fraction (≥55%), but no relations were found in those with reduced ejection fraction (<55%) [106]. This is because restrictive pattern of E/A and DT; E/A increase and DT decrease with elevation of LV endo-diastolic pressure.

Several studies demonstrated that CAVI positively correlates with the index of LV filling pressure measured by tissue Doppler echocardiography (E/E’) [107,108]. The relationship becomes stronger in patients with hypothyroidism, a high-risk group of HFpEF [109]. We reported positive linear correlation between CAVI and E/E’ in all of the enrolled patients with CVD (*n* = 100, mean age: 70 ± 8 years old) and a subgroup of patients with HFpEF (*n* = 30, mean age: 71 ± 7 years old) [110]. CAVI also positively correlated with LAVI assessed by MDCT in young adults with suspected coronary artery disease [111].

Additionally, there are previous studies evaluating diastolic dysfunction with ultrasonic strain imaging, although it is not currently included in the criteria of diastolic dysfunction [112,113,114]. LV global longitudinal diastolic strain rate measurements during the isovolumic relaxation period and during early diastole have significant association with the time constant of LV relaxation (t) measured by cardiac catheterization [112]. The index of LV diastolic function in strain imaging is less influenced by preload, whole heart motion, and tethering. A previous study demonstrated that CAVI positively correlated diastolic strain rate at the endocardial sites of posterior and the inferior wall [115].

## 8. CAVI and Cardiovascular Risk Factors

In previous cross-sectional studies, CAVI is associated with various cardiovascular risk factors. CAVI correlates with high-sensitivity cardiac troponin T [116], renal function [117], non-dipper type hypertension [118], hyperuricemia [119], dyslipidemia [120], coronary calcium [121,122], polyneuropathy in type 2 diabetics [123], paroxysmal atrial fibrillation [124], carotid intima-media [125,126,127,128], and lacunar silent cerebral infarction in patients with atrial fibrillation [129]. In addition, CAVI correlates with skin autofluorescence, which evaluates deposition of advanced glycation end products by detecting light intensity per nanometer between 300 and 420 nm in patients with type 2 diabetes mellitus [130]. 

CAVI is significantly lower in physically active women doing aerobic exercise in comparison with controls with similar ages [131]. However, for detecting peripheral arterial stiffness in younger-age patients with Kawasaki disease, a report suggested that baPWV is superior to CAVI [132]. Interestingly, there are several studies suggesting the difference of mean CAVI among countries [133,134,135]. The difference may be attributed to predisposing condition of atherosclerosis such as diet, lifestyles, and gene polymorphism [136].

## 9. CAVI and Adverse Cardiovascular Events

To date, there are increasing numbers of studies suggesting the prognostic value of CAVI for cardiovascular events. But most of them are single-centered and the sample size are small. The studies targeted mainly thromboembolic events for the endpoints such as traditional major adverse cardiac events (MACE) including cardiac death myocardial infarction and stroke [77,137,138,139]. Recent meta-analysis of Asian population ensured CAVI is an independent CVD risk factor [140]. The cut-off values of CAVI were settled in the Japan Society for Vascular Failure; CAVI: <8 for normal, ≥ 8 and <9 for borderline, and ≥ 9 for abnormal [141,142]. 

Hemodialysis (HD) patients are characterized as severe arterial stiffness by various causes including hypertension, fluid overload, and arterial calcification due to secondary hyperparathyroidism. In patients with end-stage chronic kidney disease who underwent surgical construction of arteriovenous shunt for HD, CAVI reflects the histological arterial fibrosis in the arterial fragments [143]. However, there are few studies examining the predictive value of CAVI for event-free survival in HD patients. At present, baPWV is considered to be superior to CAVI for this purpose [144].

## 10. CAVI and Prognosis of Heart Failure

There are more obvious causal relationships between arterial stiffness and heart failure because decrease of arterial wall compliance increases cardiac afterload and exacerbate the condition of heart failure. There are the evidences that arterial stiffness assessed by baPWV [100,145] or cfPWV [146,147,148] predicts the risk of re-admission in patients with heart failure. baPWV moderately correlates with diastolic dysfunction [149] and pulmonary vascular resistance in HFpEF patients [150]. A recent study showed J-shaped association between baPWV and heart failure-related events in HFpEF patients without peripheral artery disease [151]. The incident of heart failure-related events also increased with baPWV in patients with heart failure with reduced ejection fraction [152].

CAVI is independently associated with brain natriuretic peptide levels in patients with hypertension [153]. However, there are few studies of CAVI setting hospitalization for heart failure as the endpoint. Chronic obstructive pulmonary disease (COPD) is known to often coexist in patients with heart failure and is considered to be a risk of adverse outcomes in patients with heart failure. CAVI is negatively associated with forced expiratory volume at 1.0 s in patients with heart failure [154,155]. Furthermore, CAVI increases with the grades of severity of COPD in symptomatic patients of heart failure [156]. However, the study did not demonstrate the predictive value of CAVI for re-hospitalization due to worsening heart failure nor cardiac death. However, in another study investigating elderly outpatients with chronic heart failure but no history of hospital admission due to heart failure, CAVI (≥10) was a predictor of initial hospital admission for heart failure [157]. The study shows that CAVI is associated with derivatives of reactive oxygen metabolites (d-ROMs), an oxidative stress test measuring serum hydroperoxide. The relationship of d-ROMs with worsening of heart failure is reported previously [158]. 

## 11. Future Perspective

The prospective multicenter cohort study to evaluate the usefulness of the CAVI to predict cardiovascular events in Japan (CAVI-J) is ongoing. The primary endpoints are cardiovascular death, nonfatal myocardial infarction, and stroke, and the secondary endpoints are composite cardiovascular events including all cause death, heart failure requiring hospitalization, angina pectoris with revascularization, new incidence of peripheral artery disease, abdominal aortic aneurysm, aortic dissection, and deterioration in renal function [159] The large sample-size study may strengthen the effectiveness of CAVI for predicting cardiovascular events and heart failure, and enables a comparison of the effectiveness to other modalities. 

Furthermore, we recently published a paper of a study investigating the association of CAVI with endothelial insulin resistance. Freshly isolated arterial endothelial cells (FIECs) were harvested from radial catheter sheath, a disposable device for coronary angiography. Endothelial insulin resistance was defined as a decrease of insulin-stimulated phosphorylation of eNOS at serine 1177 (p-eNOS Ser1177). CAVI negatively correlated with percent increase of insulin-mediated p-eNOS Ser1177 in the FIECs. The result suggests that high CAVI is associated with deteriorated signal transduction of IRS1-PI3K-Akt-eNOS pathway indicating endothelial dysfunction [160]. Thus, we believe that translational research linking the molecular mechanisms of arterial stiffness to the physiological vascular function test of CAVI will be more common in the future.

## 12. Conclusions

Arterial stiffness is a factor of arteriosclerosis and heart failure, especially in HFpEF. Endothelial dysfunction involves the progression of arterial stiffness for which various clinical and molecular biological research has been conducted. CAVI is a convenient yet effective marker of this disorder. The efficacy will be clarified to a greater and necessary extent in future clinical research.

## Figures and Tables

**Figure 1 ijms-20-03664-f001:**
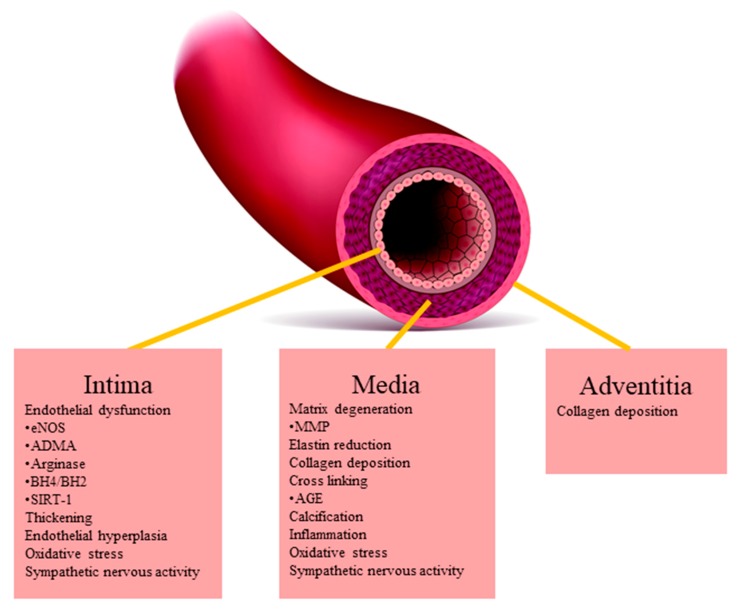
Mechanism of arterial stiffness in each layer of arterial wall. The compound and structural changes caused by mechanical wall stress contribute to arterial stiffness. Additionally, endothelial dysfunction cause vasoconstriction and is a functional cause of arterial stiffness. Oxidative stress, inflammation, and traditional cardiovascular risk factors accelerate the progression. eNOS: endothelial nitric oxide synthesis, L-ADMA: L-asymmetric dimethylarginine, SIRT-1: Sirtuin-1, BH: tetrahydrobiopterin, MMP: matrix-metalloproteinases, AGE: advanced glycation end product.

**Figure 2 ijms-20-03664-f002:**
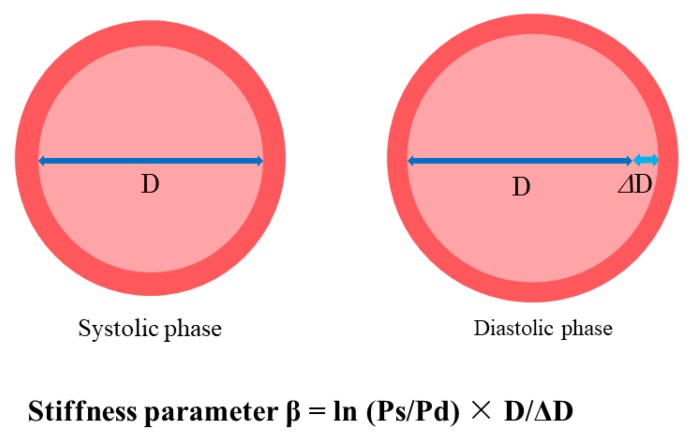
Principle of stiffness parameter β. Stiffness parameter β represents arterial distensibility, which is derived from the measurements of diameters in one section. This index was originally used for cervical and carotid arteries. Ps: systolic blood pressure, Pd: diastolic blood pressure, D: diameter of the artery, ΔD: change in diameter.

**Figure 3 ijms-20-03664-f003:**
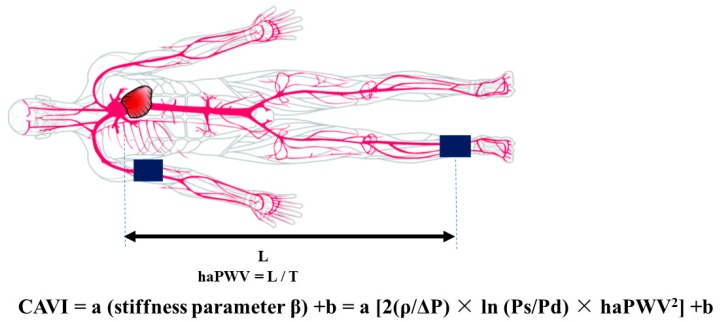
Principles of the cardio-ankle vascular index (CAVI). CAVI is an index derived from arterial intraluminal volumetric change by combining stiffness parameter β and Bramwell-Hill formula. Heart-ankle pulse wave velocity (haPWV) is obtained by dividing L (the length from the aorta to the ankle) by T (the time for the pulse wave to propagate from the aortic valve to the ankle). CAVI: cardio-ankle vascular index, haPWV: heart-ankle pulse wave velocity, Ps: systolic blood pressure, Pd: diastolic blood pressure, ΔP is pulse pressure (Ps-Pd), ρ: blood density of 1.05 g/mL. a and b are constants.

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
