# Peer review of "Arterial Stiffness Assessed by Cardio-Ankle Vascular Index"

_ijms, 2019, doi:10.3390/ijms20153664_

Round 1
Reviewer 1 Report
This is a well compiled and discussed selection of information on the utility of CAVI in assessing vascular stiffness. The authors discuss the mechanisms that cause arterial stiffness, the principles of CAVI as a technique and it's utility in different clinical circumstances. The flow of the article is nice and should promote reader engagement. One small suggestion would be to include an overview of if and where (speciality or region) CAVI is currently preferred over more common indices of arterial stiffness PWV etc.
Author Response
Response: Thank you very much for your reviewing.
We found that the comments of reviewers #1, #2 are common in regard to the necessity of description about strengths and limitation of CAVI.
Therefore, we newly created the paragraph “strengths and limitation” of CAVI”.
However, we cannot find any large-scale studies which directly compare the clinical results assessed by CAVI and PWVs. It has not been determined which is good. This is a great interest in readers, but we can only describe in the paragraph “future prospective” that a large-scale study is ongoing.
Manuscript Changes:
l We reconstructed the paragraphs. The advantage and application of CAVI were summarized in the paragraph“strengths and limitation” (Line 236-261). We described evidences that CAVI is independent from blood pressure at measurements compared to other indexes.
Other improvements:
l “precioulsry” was a minor mistake. I replaced it with “previously”. Thank you very much for detecting. (Line 320)
l I attached sentences to explain the current application of stiffness parameter β (Line 183-186)
l We added the references that CAVI correlates stiffness parameter β, which is an original principle of CAVI. (Line 233-235)
l We added an evidence which use human specimens. (Line 365-367)
l We improved the detail of mechanisms how arterial stiffness leads to organ damages.(Line 271-293)
l We changed the conclusion to match the abstract and introduction.(Line 411-414)
Reviewer 2 Report
The Authors submitted a comprehensive review on an interesting topic.
However, they could add a separate summary paragraph on “strengths and limitation” of CAVI in comparison with other expression/assessment of arterial stiffness, in particular on carotid-femoral pulse wave velocity (the gold standard of non-invasive measurement of arterial stiffness), but also on brachial-ankle pulse wave velocity and on central hemodynamics (augmentation index, central systolic blood pressure, central pulse pressure, SEVR).
In addition, the Authors could also describe a comparison with the flow mediated dilation (FMD).
Other comments:
- In the description of the evidence, PWV should be better specified (carotid-femoral, brachial-ankle, etc…?)
- Check typo-errors (e.g. line 298)
Author Response
Response: Thank you very much for your thoughtful comments.
We agreed the necessity to summarize “strengths and limitation” of CAVI”.
And also, there is a minimum necessity to explain other examinations before discussing the superiority of CAVI to them. For the purpose, we added another paragraph “physiological examination of arterial stiffness”. FMD was described in the paragraph. However, we would like to trust other articles to refer to details and clinical results of the examinations (PWVs, central blood pressure, central pulse pressure, SEVR).
Unfortunately, any large-scale studies directly compare CAVI and PWVs in clinical results cannot be found. We though to wait further large scale studies to discuss this topic.
Manuscript Changes:
l We gathered the sentences of advantage and application in for CAVI into the new paragraph “strengths and limitation” (Line 236-261). We described evidences that CAVI is independent from blood pressure at measurements compared to other indexes.
l We created a new paragraph “physiological examination of arterial stiffness”. (Line 140-172)
l We added the sentence “, and enables a comparison of the effectiveness to other modalities.” (Line 399-400)
Other improvements:
l “precioulsry” was a minor mistake. I replaced it with “previously”. Thank you very much for detecting. (Line 320)
l I attached sentences to explain the current application of stiffness parameter β (Line 183-186)
l We added the references that CAVI correlates stiffness parameter β, which is an original principle of CAVI. (Line 233-235)
l We added an evidence which use human specimens. (Line 365-367)
l We improved the detail of mechanisms how arterial stiffness leads to organ damages.(Line 271-293)
l We changed the conclusion to match the abstract and introduction.(Line 411-414)